# Evaluation of the Potency of the Pertussis Vaccine in Experimental Infection Model with *Bordetella pertussis*: Study of the Case of the Pertussis Vaccine Used in the Expanded Vaccination Program in Algeria

**DOI:** 10.3390/vaccines10060906

**Published:** 2022-06-06

**Authors:** Khedidja Tahar djebbar, Mounia Allouache, Salim Kezzal, Fouzia Benguerguoura, Chafia TouilBoukoffa, Ines Zidi, Rachida Raache, Hadda-Imene Ouzari

**Affiliations:** 1Laboratoire de Contrôle de Qualité des Vaccins et Sérums, Département de Contrôle des Produits Biologiques, Institut Pasteur d’Algérie, Route du Petit Staouel, Dely-Brahim 16047, Algeria; scienceavenir@gmail.com (M.A.); skezzal@hotmail.com (S.K.); fbenguergoura@pasteur.dz (F.B.); 2Laboratoire de Biologie Cellulaire et Moléculaire-Equipe Cytokines et NO Synthase, Faculté des Sciences Biologiques, Université des Sciences et de la TechnologieHouariBoumedienne (USTHB), Bab Ezzouar 16111, Algeria; raache_ipa@yahoo.fr; 3Laboratoire des Microorganismes et Biomolécules Actives (LR03ES03), Faculté des Sciences de Tunis (FST), Université Tunis El Manar, Campus Universitaire, Tunis 2092, Tunisia; ines.zidi@istmt.utm.tn (I.Z.); imene.ouzari@fst.utm.tn (H.-I.O.); 4Département D’Immunologie, Institut Pasteur d’Algérie, Route du Petit Staoueli, Dely-Brahim 16047, Algeria

**Keywords:** *Bordetella pertussis*, LD50, Kendrick test, relative potency, whole-cell pertussis vaccine, Algeria

## Abstract

In Algeria, vaccination against pertussis is carried out using the whole-cell pertussis vaccine combined with the diphtheria and tetanus toxoids (DTwp). The quality control of vaccines locally produced or imported is carried out before the batch release. The aim of our work was to evaluate the potency of pertussis vaccines. In the present study, five consecutive trials of potency were conducted on samples of the same batch of (DTwp) using the mouse protection test (MPT) against experimental infection of *Bordetella pertussis* strain 18323, based on the Kendrick test. The virulence of *B. pertussis strain* 18–323 was verified by the mortality of mice, with an average LD50 of 338.92, as well as the dose of the lethal test containing a mean number of LD50 of 324.43. The (MPT) test recorded a relative potency of 8.02 IU/human dose, with 95% CL of (3.56–18.05) IU/human dose. The development of the (MPT) at the laboratory of quality control of vaccines and sera at the Pasteur Institute of Algeria was effective in evaluating the potency of whole-cell pertussis vaccines. Interestingly, our study indicates that this potency is necessary for the vaccine quality assurance. Further validation is needed to strengthen the application and routine use of the test.

## 1. Introduction

Pertussis is a strictly human respiratory tract disease and is highly contagious, caused by the bacterium *Bordetella pertussis*, which is characterized by the secretion of several adhesins and toxins [1,2,3]. In recent years, the resurgence of Pertussis remains a public health problem [4,5,6].

The introduction of pertussis vaccines has strongly reduced the incidence of morbidity and mortality of the disease, particularly in infants and young children [7,8,9]. Currently, two types of vaccines are available, whole-cell vaccines and acellular vaccines (DTaP) are obtained from purified bacterial components. All are combined with diphtheria and tetanus toxoids and adsorbed on aluminum or calcium [10,11], some of them are also combined with other vaccines such as poliomyelitis, hepatitis and Haemophilusinfluenzae type B [12,13].

In Algeria, and since the introduction of vaccination strategy in 1969, it is the whole-cell pertussis vaccine, the first-generation vaccine, which is used in the expanded program of immunization [14,15]. It is characterized by high reactogenicity and better and more durable immunity. Acellular pertussis vaccines, known as second-generation vaccines, are not marketed in Algeria due to their high cost (despite their low reactogenicity and good immunogenicity) [16,17].

According to Algerian regulations, the release of a batch of vaccines is only carried outafter quality control by the competent services. Each batch of vaccine is submitted to numerous tests in order to evaluate the activity, toxicity, sterility, bacterial concentration, and certain physico-chemical parameters [15]. In this sense, the aim of our study was to develop the potency control of pertussis vaccines by the Kendrick test, based on the protection of mice against experimental infection of B. pertussis by the administered vaccine. This activity is determined by comparing with a calibrated reference pertussis vaccine against the international standard for pertussis vaccine (it is necessary to assess the titer of the pertussis component in the DTP vaccine) [18].

The test involves immunizing mice with a series of dilutions of the reference vaccine and the test vaccine. Two weeks after immunization, mice are challenged intracerebrally with a dose of a suspension of *B. pertussis* 18–323 [19]. Lethality is reported two weeks after the challenge and surviving mice are recorded.

The potency of the test vaccine was evaluated in terms of IU of standard vaccine, by the statistical method of probit analysis, using the parallel line analysis model of quantal responses, after the LD50 value determination of the *B. pertussis* challenge dose [20].

This preliminary assessment study was conducted with the aim to shed light on the application of the requirements and the validity criteria of the test. In particular, the preparation of the *B. pertussis* challenge strain and verification of its virulence (LD50), the culture conditions, the preparation of the immunization doses of the vaccines to be evaluated (ED50) and especially the choice of the mouse strain, the age, the sex, and the weight. These factors are very important for this kind of biological analysis.

## 2. Materials and Methods

### 2.1. Mice

Healthy OF1 strain mice (Charle River), weighing 14–16 g, of the different sex, equally distributed, were caged randomly in six groups of 24 mice. Three groups were used for each vaccine (3 groups for the sample vaccine and 3 groups for the standard vaccine) and one group of 10 mice for negative control, for which, the mice were injected with the vaccine diluent). Concomitantly, 5 additional groups of 10 mice were used to control the virulence of the challenge strain *B. pertussis*, including one group for the negative control (mice were injected with tryptone salt diluent). All mice were placed under conventional conditions (temperature 20–25 °C, humidity 50%) in the animal house of the quality control laboratory of the Pasteur Institute of Algeria (IPA).

### 2.2. Vaccines

Five consecutive trials were conducted on samples of the same batch of adsorbed pertussis vaccine (16 OU/human dose), in combination with diphtheria and tetanus antigens DTP (Test Vaccine 1, 2, 3, 4, and 5). In parallel, a freeze-dried reference pertussis vaccine RWS 01/11(Serum Institute of India SII), with a titer value of 63 IU/ampoule was used, and diluted to 0.5 IU/human dose of 0.5 mL. The samples were taken from the stock destined for the expanded program of vaccination.

### 2.3. Bordetella PertussisStrain 18–323 and Growing Conditions

The *Bordetella pertussis* strain 18–323 used in our study, is supplied by SII (Serum Institute of India), it was previously prepared, re-suspended, aliquoted, and preserved in suspension in liquid nitrogen. It was tested and validated for its virulence in non-immune mice. The bacteria were cultured at 37 °C, on a Bordet-Gengou agar medium (Becton Dickinson, Franklin Lakes, NJ, USA), supplemented with 15% of defibrinated fresh horse blood. The purity, morphology and characteristics of the strain were verified by Gram staining, oxidase and catalase testing, and by using the API 20 NE system (bio Merieux, Marcy L’Etoile, France). The identity was confirmed by the agglutination test using the specific antiserum Fim 2 and Fim 3 (SII), according to the WHO manual [20].

### 2.4. Mouse Protection Test (MPT)

The MPT was carried out on the basis of WHO recommendations (WHO, [21]), according to the following steps (Figure 1):

#### 2.4.1. Immunization

Three serial dilutions were prepared in sterile saline solution by a dilution factor of 5 of the tested vaccine respectively (1/8, 1/40 and 1/200) of the 16 OU human dose equal to 2, 0.4, and 0.08 OU/human dose). Three other dilutions of the standard vaccine were made respectively (1, 1/5 and 1/25 of 0.5 IU/human dose equal to 0.5, 0.1, and 0.02 IU/human dose. A batch of 24 mice was immunized intraperitoneally with 0.5 mL of each dilution. In parallel, a control group of 10 mice received 0.5 mL of sterile saline. The mice were observed for 14 to 17 days.

#### 2.4.2. Preparation of *B. pertussis* 18–323 Challenge Suspension

The *B. pertussis* 18–323 suspension of the Challenge test was prepared in 1% solution of Casamino Acid (Difco, PH: 7.0–7.2), from a culture incubated for 24 h at 37 °C on abordet-gengou agar with 15% defibrinated blood. The bacterial suspension was adjusted to 10 OU/mL (10 × 10^9^ organisms/mL), based on the fifth international standard of opacity by visual comparison. This suspension was diluted to 1:3000 with 1% Casamino acid diluent, which is the challenge suspension. Further dilutions of the challenge suspension were prepared for the estimation of the virulence of the challenge culture (LD50), and bacterial count determination (viability of challenge suspension of *B. pertussis*).

#### 2.4.3. Injection of Challenge Suspension of *B. pertussis* (Challenge Test)

After 15 days of observation, immunized mice with the sample and standard vaccine were infected intracerebrally with bacterial suspension (100,000 bacteria/0.03 mL) of the *B. pertussis.* The time between the preparation of the challenge dose and its injection into the brain did not exceed 2 and 1/2 h. In the same way, and in parallel, a control group of 40 mice, divided into four subgroups of 10 mice, was injected intracerebrally with 10,000, 2000, 400, 80 bacteria/dose of 0.03 mL, respectively. A negative control group received 1% dilute casamino acid. The mice were observed daily for 14 days. Death before day 3 was not considered significant (traumatic death). Immediately after preparation of challenge dilution, 0.2 mL of the last dose (80 bacteria/0.03 mL) used in the control group of the *B. pertussis* virulence test, was plated on a Bordet-Gengou agar with 15% defibrinated blood, and was incubated at 37 °C for four to five days to determine the number of unit forming colonies in the Challenge suspension.

The results were analyzed statistically to estimate the virulence of the *B. pertussis* challenge suspension (LD50), and to assess the effective dose (ED50) and the potency of the tested vaccine.

### 2.5. Statistical Analysis

Statistical significance tests were carried out at the 5% probability level.

Data from the non-immunized control group of mice infected with *B. pertussis* strain 18–323 were used to evaluate the LD50 value of the *B. pertussis* 18–323 test suspension by the statistical method of Probit analysis, relating the proportions of dead mice to the log concentration of each dilution of the test suspension. The LD50 and the number of estimated LD50s were expressed as the mean along with their 95% confidence limit and standard deviation, from the five trials. Statistical analysis was performed by SPSS version 23 software (Statistical Package for the Social Sciences software, Version 23, SPSS Chicago (IL), USA).

Data from the group of immunized mice and challenged with *B. pertussis* 18–323 were used to evaluate the Median Immunizing Dose ED50 and the relative potency of the test vaccine by the statistical method of probit analysis. This method is based on the parallel line analysis model of quantal responses, relating the probit of the proportion of surviving immunized mice after the challenge to the logarithm of the dose of each vaccine preparation, using a WHO Software Combistats (version 4). The data were adjusted using linear regression. The results are expressed as IU/human dose.

The trials were analyzed using a normal ANOVA to test the consistency of the data with the chosen model. The validity of the individual tests was assessed in terms of total deviations from the linear model and deviations from linearity and parallelism of the lines by comparing two preparations (test vaccine and standard vaccine).

The statistically validated relative potencies were combined and tested for homogeneity, using the chi-squared test. Homogeneous validated estimates were combined as weighted geometric means (*p* ≥ 0.05). Validated estimations that showed significant heterogeneity (*p* ≤ 0.05) were combined as unweighted geometric means.

### 2.6. Validity Criteria

(1) At least 87.5% of mice immunized with each dilution of the standard and experimental vaccines should survive after the 14-day immunization period [22]. (2) The trial is validated if the titration of the *B. pertussis* suspension indicates that the test dose of 0.03 mL will contain 100 and 1000 LD50, and the number of colony-forming units CFU in 1 LD50 must not be more than 300 colonies and the number of CFU in the last dose of challenge suspension must not be less than 10. (3) The assay is validated if the ED50 of each vaccine was between the lowest and highest dose used to immunize the mice, and if the probit regression of the proportion of mice protected as a function of the logarithm of the dose of each vaccine preparation was significant (*p* ≤ 0.05). (4) Analysis of the statistical tests indicates that each of these trials is statistically valid, as long as they show no significant deviations from linearity and parallelism (*p* ≥ 0.05). (5) Any trial for which deviations from linearity or parallelism were significant (*p* < 0.05), and for which total deviations from the model were also significant (*p*< 0.05) was considered invalid. The activity estimates for these trials were not included in the calculated mean estimates. (6) The vaccine has successfully passed the test when the measured potency is at least 4 IU/human dose and if the lower 95% fiducial limit (confidence limit) is not less than 2 IU/human dose (WHO, European Pharmacopoeia).

### 2.7. Ethical Considerations

The study was conducted in strict accordance with the ethical care and use of laboratory animals. All efforts were made to minimize the number of animals used and their suffering. All procedures performed on animals were in accordance with the ethical standards of the Directive of the European Parliament and of the Council on the protection of animals used for scientific purposes (2010/63/EU). All animal experiments carried out in vivo approved by the Research Ethics Committee of the Institute Pasteur of Algeria, and also approved by the Algerian Association for the Sciences of Animal Experimentation (AAAES).

## 3. Results

### 3.1. Identity and Viability of B. pertussis 18–323

The *B. pertussis* strain 18–323 presented all of the typical characteristics of the *B. pertussis* species. The colonies were white opaque, smooth, in phase I. The gram stain showed gram negative coccobacilli, the oxidase, catalase and agglutination tests with the specific antiserum Fim2 and Fim3 revealed a positive reaction.

The viability of the infecting dose prior to the challenge was verified by growth and colony counts of the last dose used in the control group (80 microorganisms/0.2 mL) on Bordet Gengou medium supplemented with horse blood. The colony count ranged from 13 to 33 colony forming units (CFU), corresponding to 2.43 × 10^3^ to 6.18 × 10^3^ viable particles/dose of challenge) (Table 1).

### 3.2. Infection by B. pertusis Strain 18–323

Four days after inoculation with *B. pertussis*, a clinical picture was established in the control group. This group was characterised by the typical symptoms of pertussis. The symptoms were revealed even with the low doses of B. pertussis injected, which appeared later than those injected with the higher doses. Symptoms included an accelerated heart rate with respiration trouble, reduced activity, slow gait, cyanotic hind legs and progressive loss of muscle control. Death occurred on the fifthday for some mice.

### 3.3. Determination of the Lethal Dose of B. pertussis 18–323 (LD50) and the Number of LD50 Used in Each Trial

Despite the precision taken during the adjustment of the bacterial suspension, in the five trials, the estimated LD50 of the challenge dose of *B. pertussis* injected into non-immunized control mice varied from 202.17 to 463.20 with a mean of 338.92 ± 111.60. The number of the LD50s used in the challenge dose varied considerably from 215.88 to 494.61 times LD50 with a mean of 324.43 ± 114.68, in conformity with the WHO standard and the European Pharmacopoeia, which have recommended an interval of 100 to 1000 times LD50 (Table 1).

In addition, In the control group of each trial, statistical analysis showed that the curve of dose-response is dependent on the dose of *B. pertussis* (the curve of the proportion of dead mice transformed into a probit). Moreover, the curve is not significantly deviated from linearity (*p* ≥ 0.05). The slope was significantly different from zero (*p* ≤ 0.05). We observed with interest a strong relationship (R^2^ = 0.99 to 1) between mouse mortality and the concentrations of *B. pertussis* used in the control group.

### 3.4. Estimation of the Relative Potency and the Protective Dose ED50 of the Pertussis Componentof the DTP Test Vaccine against B. pertussis Strain 18–323 Infection

After 14 days of immunization, all 5 trials recorded an immunization rate above 87.5%, except for the first trial, which registered a decrease in the number of immunized mice for the highest dilution of the standard vaccine (79%), and the test vaccine (75%). Fourteen days after challenge of the immunized mice, the percentage of survival ranged from 70% to 95% for the highest doses, and from 5% to 22% for the lowest doses in the chosen range. These results are consistent with this type of bioassay, aiming to achieve a 50% survival rate (Table 2).

The ED50 protective dose of the standard and test vaccines was ranged from the lowest to the highest dose recommended for each trial. It varied from 0.08 to 0.40 IU/human dose (Table 3).

The analysis of variance, calculated by the probit analysis program, revealed that all trials indicated a highly significant regression of the dose response curve (the curve of the proportion of surviving mice transformed in probit as a function of the dose of each vaccine preparation) (Table 3). In addition, they showed no significant deviations from linearity and parallelism (*p* ˃ 0.05) with exception of the trial 4. In this case, the statistical analysis revealed a significant deviation from parallelism (*p* ˂ 0.05). Again, the common slope of the dose-response regression curve of the proportion of surviving mice transformed to a probit as a function of the dose of each vaccine preparation in each trial was determined. It varied slightly from 0.30 to 0.54, and showed no significant difference between the prepared dose of each vaccine for each trial (*p* ˃ 0.05) (Figure 2).

The relative potency was determined for each trial, along with the 95% confidence limits. Trial 1 satisfied the specifications for relative potency 9.77 IU/human dose (2.15–93.70), despite recording a decrease in the number of mice after immunization for the higher dilution of the two preparations. Trials 2 and 3 recorded potency less than 4 IU/dose and the lower confidence limit was less than 2 IU/human dose, and trial 4 showed potency greater than 4 IU/dose, but the lower confidence limit was also less than 2 IU/human dose. Trial 5 satisfied the relative potency specification. However, it was greater than 4 IU/dose (7.67 IU/human dose) and the lower limit of the 95% fiducial limits was greater than 2 IU/human dose, (3.31–19.96) (Table 3).

The two valid relative potency estimates from trial 1 and trial 5 were combined and tested for homogeneity. However, the chi-2 test showed significant consistency between the two estimates (*p* = 0.82) and were considered homogeneous. The two values were combined as a weighted geometric mean of 8.02 IU/human dose with their fiducial limits (3.56–18.05) (Table 4).

## 4. Discussion

Whole-cell pertussis vaccines are still used in Algeria and induced protective and long-lasting immunity despite their reactogenic effects [23]. The potency test is one of the important tests to guarantee the quality of the vaccine in terms of efficacy and safety [24]. It is based on the measurement of one or more parameters ensuring that the antigen has the immune response induction by the antigen. Indeed, the intracerebral mouse protection test (MPT.Ic) or Kendrick’s test was the subject of our study, in order to develop the activity control of monovalent pertussis vaccines or associated with other antigens, and marketed in Algeria. However, and despite the variability that characterizes the mouse protection test (MPT), and the development of several alternative methods, such as the serological potency assay [25,26,27,28], the intra-nasal respiratory challenge test [29,30], and the nitric oxide and hydrogen superoxide assay [31,32], the (MPT) test remains the official potency test and the reference method for assessing the potency of pertussis vaccines [24]. In addition, it was the only test that shows a relationship with protection against pertussis in children in clinical trials [33]. Furthermore, the test is performed on mice, as an animal model, which is available and easy to maintain and breed, unlike other associated components such as diphtheria and tetanus antigen, which require the use of guinea pigs, whose maintenance requires a lot of efforts and means. It therefore appeared necessary to develop the potency test for pertussis vaccines. Of note, these vaccines are available and marketed in Algeria in the form of vaccines combined with diphtheria and tetanus vaccines.

In the present study, five trials of potency were performed on an imported batch of a combined diphtheria-tetanus-pertussis (DTP) vaccine. The results obtained have always shown the variability reported in this type of biological test. The observation of this variability was coherent with previous studies [19,34,35,36]. Indeed, the mouse protection test against experimental *B. pertussis* infection by pertussis vaccine is influenced both by the immunization dose ED50, which corresponds to the efficacy of the vaccine [19,22,37], the virulence of the challenge strain (LD50) of *B. pertussis* [38] and the strain of mouse used [24,36,39]. All of these factors constitute a source of variability that affects the potency test.

In our study, the lethal dose 50 (LD50) and the value of LD50 of *B. pertussis* were evaluated in each potency test, according to the recommendations of the WHO and the European Pharmacopoeia for this test. The results showed that the *B. pertussis* strain is a Phase I virulence strain, and the virulence factors were expressed, *in vivo*, by a clinical picture characteristic of pertussis and mortality of mice from the fifth day onwards for the higher doses of the strain tested. In addition, the estimated LD50 number was within the range recommended by the WHO standard (100 to 1000 LD50), and the number of viable particles per LD50 was close to 300 colony-forming units with some variation.These criteria showed compliance with all steps of the culture, preparation and adjustment of the bacterial suspension. However, the slight variation recorded in the number of viable organisms of *B. pertussis* in a LD50 (338.92 ± 111.60), and the number of LD50s of the challenge suspension (324.43 ± 114.68), of the five trials, is essentially explained by the culture conditions of the strain, citing the age of the culture and the quality of culture media used [40,41]. These results are similar to those reported by several other studies [33,35,38]. Also adding that the LD50 could be subject to variations during intracerebral injection, or there could be a loss of some of the bacterial inoculum following the use of a rough-tipped needle [19,21].

In addition, the number of viable *B. pertussis* organisms has an influence on the virulence of the challenge strain. In this sense, the viability of *B. pertussis* was checked by plating and counting the last dilution of the challenge suspension, prepared for the LD50 titration, on Bordet and Gengou medium. The vitality of *B. pertussis* was positive, but the number of colonies forming units was much lower than the adjusted starting number in the challenge suspension. This decrease in CFU count was explained by the poor growth conditions of *B. pertussis* on Bordet and Gengou medium, which contains peptone. These observations are consistent with previous studies [15,19,41,42]. According to Xing [24], a culture of 72 h will decrease by a proportion of viable organisms of 14% or much less, and a *B. pertussis* strain grown for too long and may lose the phase I virulence factors [43,44,45].

The Kendrick test also depends on the median immunization dose ED50 of the vaccine tested. 50% of mice survive after inoculation with *B. pertussis* at this vaccine dose. However, the efficacy of the vaccine (ED50), can be determined only if the immunized mice induce a measurable response following the challenge test. This response is more than 0% and less than 100% of mice survival [46]. The reason why the effective dose range of the vaccine is limited is that it is defined as three doses or dilutions, and implies that the ED50 must be within the range of the three doses of used vaccine [20]. Indeed, these criteria are consistent with our results. However, the calculated ED50 of the standard and test vaccines in the five trials were within the range of the three-dose range chosen, and for which the data showed a percentage survival, which was 71% to 95% for the highest dose, and 5% to 22% for the lowest dose of the range used. The similar results have been reported in several studies [22,35,47]. Furthermore, in our study the ED50 calculated by the program chosen varied from 0.0810 to 0.402 IU/human dose from trial to trial. This variation was probably due to the strain of mouse used in relation to age, weight and sex, or other technical factors in the trial [24]. This shows the importance of the use of the standard vaccine. Indeed, it could be an international or internal reference standard according to vaccine activity. These data will make it possible to optimize the immunization dose for the mouse strain used [12,48], and to control the variability that characterizes this bioassay. Furthermore, the calibration of the internal references by an international reference constitutes a very important point for the consistency of the potency estimates of the tested vaccines [24]. However, in our current study, this parameter was verified by the use of the standard vaccine indicating a highly significant regression of the response of immunized mice as a function of the log transformation of the vaccine dose in all trials performed (*p* < 0.01). In the same context, it also observed that the potency assessment of the test vaccine compared to the standard vaccine of a known titer value was very important, and more reliable than the calculated ED50 value [19]. In fact, this evaluation depends on the technical variations that may be unavoidable. We will therefore be able to avoid drawing false conclusions for the standard vaccine in the case of the use of a mouse strain requiring a lower or higher immunization dose than the mice usually used [20].

Statistical analysis using the probit analysis method, based on the parallel line analysis model used for the five trials and analysis of variance, showed levels of probability associated with deviations from the model. However, each trial gave a significant regression of the response curve of the immunized mice transformed into probit as a function of the log of the vaccine dose used (*p* ≤ 0.05), and the common slope analysis revealed no significant difference between the preparations of each trial (*p* > 0.05). Similar results have been reported in several studies [22,35,38,47].

Our results showed that the parallel lines model was the most appropriate model to evaluate the potency of the test vaccine relative to the standard vaccine (relative potency). In this model, the linearity and parallelism of the dose-response regression curves were assessed, and used for the validity of each trial. However, the study of the validity criteria and the statistical study by the chosen analysis model allowed for concluding that:

-Trial 1 satisfied the validation criteria, despite a decrease in the number of mice during the immunization period by the highest dose of the interval recommended for the two vaccine preparations (standard and test vaccine). However, the analysis of variance showed that the trial was statistically validated. This was explained by the WHO recommendation that in some cases, the immunized group with the highest dose of the standard vaccine should be monitored for signs of abnormalities in gait and posture.That 50% of the mice in this group should show clinical signs [15]. In addition, this situation can also be explained by accidents or technical errors that can occur in the first trials in this type of dosage, such as the use of an incorrect injection technique by an inexperienced staff member, or incorrect storage of a product [20].-Trials 2 and 3 met all of the validity criteria for the test, except that the lower limit of the 95% confidence interval was less than 2 IU/human dose, which is an important criterion in the potency specification. This result rendered the trial invalid.-Trial 4 did not satisfy all of the criteria. However, analysis of variance showed that the dose-response regression curves for the test and standard vaccines deviated significantly from parallelism (*p* < 0.05) (Table 3). This result led to a fundamental invalidity of the trial explained by the preparation of the dilutions of the two vaccines used Although, the calculated relative potency and its lower bound were within recommended specifications and the joint slope of the dose-response regression showed no significant difference in the preparations of each vaccine (standard and test vaccine). Indeed, whatever the situation, the parallelism of the dose-response regression curves of the two vaccine preparations takes a very important consideration in the validity of the trial and shows the importance of the preparation of the dilutions of the vaccine to be tested, taking into account the titer value of the standard vaccine used.-Trial 5 met all validity criteria and the statistical analysis showed a relative potency greater than 4 IU/human dose with the lower limit of the 95% confidence limits greater than 2 IU/human dose. The dose-response regression was highly significant (*p* = 0.00) for both the standard and test vaccines, and the analysis showed no significant deviation from linearity and parallelism. Indeed, the range of doses chosen for the standard and test vaccines gave responses within the linear range, and the dilutions of both vaccine preparations were performed correctly, which explained the absence of significant deviation from linearity and parallelism.-The relative potency of the tested pertussis vaccine (DTP vaccine test) was calculated from the two valid estimates of relative potency from Trial n°1 and Trial n°5 that satisfied the relative potency specifications and test validity criteria. The two estimates were homogeneous (*p* > 0.05). This showed that the estimated relative potency is approximately normally distributed with a mean defined by the estimation of the log potency and the weight (given by the inverse of the variance of the log potency). This information can be used to determine the probability of particular values for subsequent estimates [20]. However, more several tests are necessary for further validation of the potency assessment of whole-cell pertussis vaccines.

## 5. Conclusions

Analysis of the study results showed that the intracerebral mouse protection test against experimental *B. pertussis* infection was adequate for estimating the potency of whole-cell pertussis vaccines used in the national immunization program despite the known variability in this test. However, when performing this type of bioassay, it is important to place greater emphasis on the factors that affect it, including the use of a strain of mice from the same stock, with respect to the randomization of mice for each treatment group, in order to make the results more homogeneous. In addition, the use of a virulent strain of *B. pertussis* of Phase I is important for the challenge, implying control and verification of the growth conditions and the quality of the culture media used. Moreover, the availability of reference standards of the vaccine to be tested is very important for the accurate estimation of potency.

## Figures and Tables

**Figure 1 vaccines-10-00906-f001:**
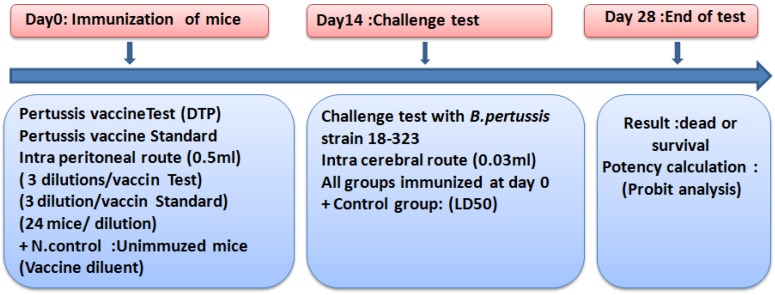
The mouse protection test steps (MPT) (Kendrick test): (1) Day 0: immunization of one group of 24 mice by one dilution, simultaneously with a whole-cell pertussis vaccine combined with diphtheria and tetanus antigens (DTP) and a standard pertussis vaccine (three dilutions/vaccine, 0.5 mL, intraperitoneally), Plus a negative control (N.control) of 10 non-immunized miceinjected with diluent vaccine. (2) Day 14: challenge test with a suspension of *B. pertussis* of all groups of mice (0.03 mL, Intracerebrally), plus a control group of non-immunized mice infected with B. pertussis (LD50 estimation). (3) Day 28: final reading and recording of surviving mice to estimate the potency relative to the pertussis vaccine tested (Probit analysis).

**Figure 2 vaccines-10-00906-f002:**
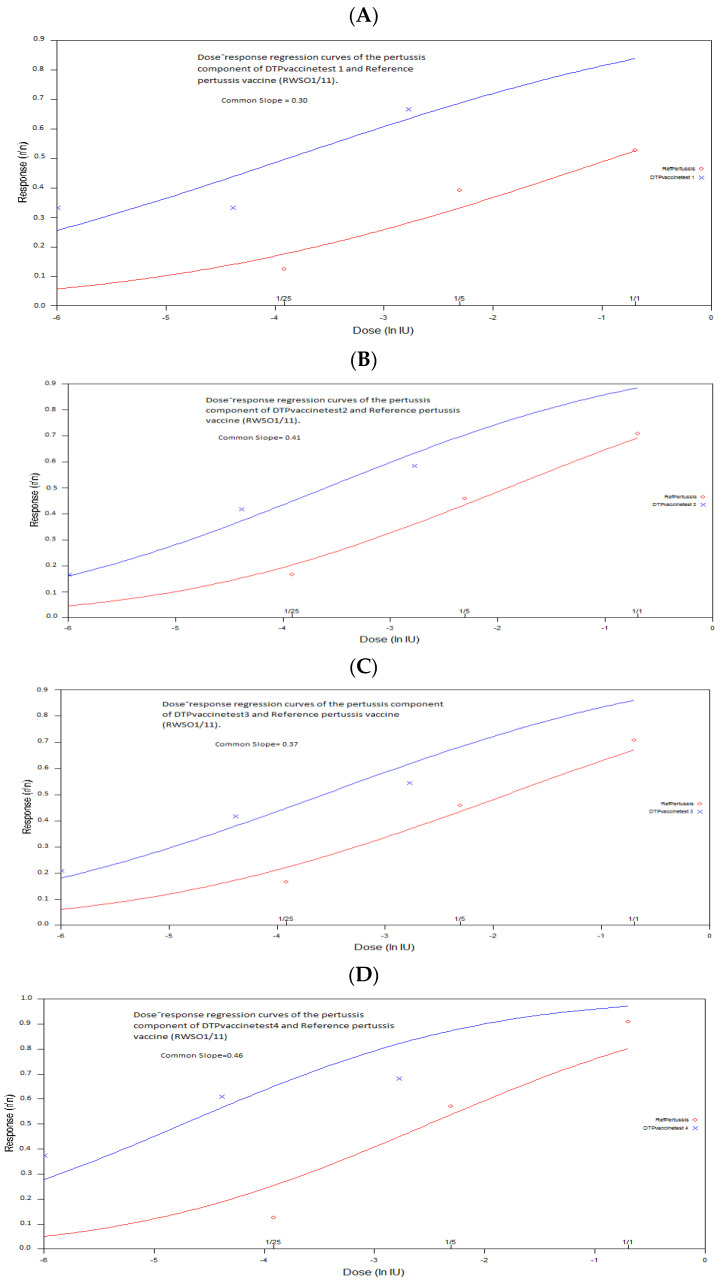
Estimation of dose response regression lines of the pertussis component of DTP vaccine test and reference pertussis vaccine (RWSO1/11) by the parallel line method using WHO combistats software(dose response regression curves of the proportion of surviving mice transformed to a probit as a function of the dose): (**A**)The pertussis component of DTP vaccine test 1 and reference pertussis vaccine (RWSO1/11),(**B**) the pertussis component of DTP vaccine test 2 and reference pertussis vaccine (RWSO1/11),(**C**) the pertussis component of DTP vaccine test 3 and reference pertussis vaccine (RWSO1/1), (**D**) the pertussis component of DTP vaccine test 4 and reference pertussis vaccine (RWSO1/11) (**E**) the pertussis component of DTPvaccine test5 and reference pertussis vaccine (RWSO1/11).

**Table 1 vaccines-10-00906-t001:** Estimation of the lethal LD50 and number of LD50 of *B. pertussis* strain 18–323 used in the challenge test of the pertussis component of the five DTP vaccine trials (vaccine test 1, 2, 3, 4, 5) in mice of control group/and Control of the viability of *B. pertussis* strain 18–323.

Trial N°	Dose B.p (µo/0.03 mL)	NB Dead Mice (Control Group)	Viability B.p Nb Colonies (M) (BG Medium)	ViabilityB.p CFU Counting/ Challenge Dose	LD50 (95% IC)	NB LD50/ Challenge Dose	Slope of Curve *p* ≤ 0.05	R^2^
1	10,000	10/10	13	2.43 × 10^3^	318.87 (144.95–690.10)	313.65	0.84	1
	2000	10/10						
	400	10/4						
	80	10/2						
	C.Diluent	10/10						
2	10,000	10/10	18	3.37 × 10^3^	269.3 (71.19–631.54)	371.32	1.2	0.88
	2000	10/8						
	400	10/7						
	80	10/2						
	C.Diluent	10/10						
3	10,000	10/9	15	2.81 × 10^3^	202.17 (7.25–662.39)	494.61	0.82	0.92
	2000	10/8						
	400	10/7						
	80	10/3						
	C.Diluent	10/10						
4	10,000	10/10	33	6.18 × 10^3^	463.20 (190.54–1018.43)	215.88	1.52	0.98
	2000	10/8						
	400	10/5						
	80	10/1						
	C.Diluent	10/10						
5	10,000	10/10	24	4.50 × 10^3^	441.06 (161.54–104530)	226.72	1.2	0.97
	2000	10/8						
	400	10/4						
	80	10/2						
	C.Diluent	10/10						
Mean 95% CI					338.92 (200.34–477.50)	324.43 (182.04–466.83)		
SD					111.6	114.68		

B.p: *Bordetella pertussis*, NB: number, C.Diluent: control diluent, B.G: Bordet and Gengou, µo: Micro-organism, R^2^: coefficient de determination, SD: standard deviation, M: mean of two plate of BG.

**Table 2 vaccines-10-00906-t002:** Mouse protection test results of the pertussis component of the DTP vaccine trials (vaccine test 1, 2, 3, 4, 5): Percentage of immunization before the challenge test and percentage survival of immunized mice after the challenge test.

		Dilution	Number of Mice after Immunization	Percentage of Immunization ^(a)^ (%)	Number of Surviving Immune Mice after the Challenge Test	Percentage of Survival ^(b)^ after the Challenge Test (%)
Trial 1	Standard Vaccine P ^*^ 0.5 IU/dose	1:1	19/24	**76.16**	10/19	70.96
		1:5	23/24	95.83	9/23	34.28
		1:25	24/24	100	3/24	6.38
	DTP vaccine test 1	1:8	18/24	**75**	12/18	82.35
		1:40	24/24	100	8/24	42.1
		1:200	24/24	100	8/24	17.39
	N.C	V.S	10/10		09/10 **	
Trial 2	Standard Vaccine P ^*^ 0.5 IU/dose	1:1	24/24	100	17/24	82.05
		1:5	24/24	100	11/24	42.85
		1:25	24/24	100	4/24	9.09
	DTP vaccine test 2	1:8	24/24	100	14/24	73.68
		1:40	24/24	100	10/24	36.84
		1:200	24/24	100	4/24	8.33
	N.C	V.S	10/10		10/10	
Trial 3	Standard Vaccine P * 0.5 IU/dose	1:1	24/24	100	17/24	82.05
		1:5	24/24	100	11/24	42.85
		1:25	24/24	100	4/24	9.09
	DTP vaccine test 3	1:8	22/24	91.66	12/22	69.23
		1:40	24/24	100	10/24	36.58
		1:200	24/24	100	5/24	10
	NC	V.S	10/10		10/10	
Trial 4	Standard Vaccine P * 0.5 IU/dose	1:1	22/24	91.66	20/22	94.59
		1:5	21/24	87.5	12/21	57.69
		1:25	24/24	100	3/24	17.64
	DTP vaccine test 4	1:8	22/24	9.66	15/22	84.44
		1:40	23/24	95.83	14/23	58.97
		1:200	24/24	100	9/24	22.5
	N.C	V.S	10/10		10/10	
Trial 5	Standard Vaccine P * 0.5 IU/dose	1:1	24/24	100	18/24	84.21
		1:5	24/24	100	12/24	43.75
		1:25	24/24	100	2/24	4.76
	DTP vaccine test 5	1:8	24/24	100	19/24	88.63
		1:40	23/24	95.83	15/23	60.6
		1:200	22/24	91.66	5/22	14.28
	N.C	V.S	10/10		10/10	

*: Standard vaccine pertussis, ^a^: percentage equation method, ^b^: cumulative number method (cumulative frequencies) N.C, negative control, V.S, vaccine solvent, **: one lost to negative group.

**Table 3 vaccines-10-00906-t003:** Results of the estimation of the relative potency and protective dose ED50 of the five trials performed on the pertussis component of DTP test vaccine (Vaccine test 1, 2, 3, 4,5).

Trial N°	P0TENCY (IU/Human Dose) 95% FL (*p* ≤ 0.05)	Number of ED50/Human Dose	ED50 (IU/Human Dose) 95% FI (*p* ≤ 0.05)	Common Slope B	Linearity (*p* ≥ 0.05)	Parallelism (*p* ≥ 0.05)
**1**	9.77 (2.15–93.70)	24.28	0.40 (0.44–1.35)	0.30	No significant deviation *p =* 0.32	No significant deviation *p* = 0.52
**2**	2.71 (0.79–8.18)	18.24	0.14 (0.11–0.21)	0.41	No significant deviation *p =* 0.83	No significant deviation *p =* 0.53
**3**	2.720 (0.66–9.21)	17.75	0.15 (0.12–0.22)	0.37	No significant deviation *p =* 0.87	No significant deviation *p =* 0.29
**4**	4.65 (1.67–13.46)	56.97	0.08 (0.02–0.10)	0.46	No significant deviation *p =* 0.75	significant deviation *p* = 0.006(˂0.05) No Parallelism
**5**	7.672 (3.31–19.96)	54.47	0.14 (0.11–0.18)	0.54	No significant deviation *p =* 0.33	No significant deviation *p =* 0.48

FL, fiducial limits.

**Table 4 vaccines-10-00906-t004:** Estimation of the weighted geometric mean of the relative potency of the pertussis component of DTP vaccine tested (DTP vaccine test 1 and vaccine test 5). Statistical analysis for homogeneity (*p* ≥ 0.05) was performed using the chi-squared test under WHO combistats software.

Relative Potency			
	Lower Limit	Estimate	Upper Limit
DTP vaccine test 1	2.15	9.77	93.70
DTP vaccine test 5	3.31	7.67	19.96
Weighted combination *			
potency	3.56	8.02	18.05
Estimate in percent	44.4%	100%	225.0%

*: Geometric combination DTP vaccine test 1 and 5 (*p* = 0.82).

## Data Availability

Not applicable.

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
