# Peer review of "Evaluation of the Potency of the Pertussis Vaccine in Experimental Infection Model with Bordetella pertussis: Study of the Case of the Pertussis Vaccine Used in the Expanded Vaccination Program in Algeria"

_vaccines, 2022, doi:10.3390/vaccines10060906_

Round 1

Reviewer 1 Report

This manuscript describes the mouse potency test for quality control of pertussis vaccines in Algeria. The authors performed five independent tests based on the Kendrick test, and summarized the results. They concluded that the test is adequate for estimating the potency of whole-cell pertussis vaccines.

In the Introduction section, the authors describe that “this preliminary assessment study was conducted with the aim to shed light on the application of the requirements and the validity criteria of the test, .... and especially the choice of the mouse strain, the age, the sex, the weight (lines 68-74). However, they did not investigate the factors such as vaccine doses and mouse strain, i.e., all tests were done under the same conditions. Unfortunately, I did not find useful information about the mouse potency test in the MS. As a side note, the mouse potency test (modified Kendrick test) is conducted for routine lot release of aP vaccines in some Asian countries.

Minor comments:

Title: the word “activity” is inappropriate. In the MS, the authors use both the words “activity” and “potency” for the effectiveness of pertussis vaccine. I think the “potency” is appropriate.
Line 61: the word “valence” is unclear.
Lines 104 and 110: change the “sterile physiological water at 0.9%” to the “sterile saline”.
Line 113: change to “Bordet-Gengou agar with ...”.
Line 134: DL50?
Lines 185-189: the sentences are duplicated (lines 179-183).
Line 210: the phrase “Evolution of the experimental disease” is unclear.
Table 3: the word “Pas de’ecart” is French?
1: expand the horizontal axes to -9.
Lines 310-397: the discussion should be shortened. The writing is lack conciseness.
The MS should be improved by a through English proofreading.

Author Response

Dear Reviewer1

We are very grateful to reviewer 1 for his thoughtful suggestions and recommendations, which greatly improved the quality of our manuscript. Based on your suggestions, we have made careful modifications to the manuscript. We would like to thank you for reviewing the manuscript and taking the time to improve the manuscript.

Here we present our point-by-point response to the comments.

Our study is preliminary, the goal of which is to implement a routine test in the laboratory of the quality control l for vaccines and serums at the Pasteur Institute in Algeria (IPA), relating to the activity of pertussis vaccines (Kenfrick's test). It will make it possible to assess the validity criteria. Kendrick's test is an in vivo test, characterized by a large variation. In this context, we carried out five tests on the same batch of vaccine and under the same conditions. We used mice from the same stock, as well as the same doses of vaccine administered in each trial according to the WHO protocol. With a focus on the choice of the mouse strain, the preparation of the dilutions of the immunization doses of the vaccine to be tested and in particular on the preparation of the test strain of B. pertussis as well as the verification of its virulence (LD50).

Minor comments:

Point 1 : Title: the word “activity” is inappropriate. In the MS, the authors use both the words “activity” and “potency” for the effectiveness of pertussis vaccine. I think the “potency” is appropriate.

Response1 : Thank you for your advices. It was corrected «   Activity » was replaced by potency.

Point 2 : Line 61 : the word “valence” is unclear.

Response 2 : Line 61, For clarity. The word valence is replaced by the word component. 

Point 3 : Lines 104 and 110 : change the “sterile physiological water at 0.9%” to the “sterile saline”.

Response 3 : Line 104 and 110 : Thank you for your recommendation and constructive advices. ‘Sterile physiological water at 0.9%' is replaced by sterile saline.

Point 4 : Line 113 : change to “Bordet-Gengou agar with ...”.

Response 4 : Line 113 : Thank you .It is corrected .Bordet Gengou is replaced by Border-Gengou agar with... 

 Point 5 : Line 134 : DL50 ?

Response 5 : Line 134 : For clarity, DL50 is replaced by LD50 (lethal dose median).

Point 6 : Lines 185-189 : the sentences are duplicated (lines 179-183).

Response 6 : Line 179- 183 : We acknowledge the reviewer and the duplicated sentence was deleted.

Point 7 : Line 210 : the phrase “Evolution of the experimental disease” is unclear.

Response 7 : The phrase is relative to the period of observation of symptoms and signs of pertussis in mice after the challenge by B. pertussis ( especially the control group). For clarity, the phrase was changed to “Infection by B. pertusis strain 18-323”

Point 8 : Table 3 : the word “Pas d’ecart” is French ?

Response 8 : We apologize. The word "Pas d’écard significatif" is replaced by No significant deviation.

Point 9 : 1: expand the horizontal axes to -9.

Response 9 : For clarity, The graphs are produced by the WHO combistats software, according to the vaccine doses used (the starting dilutions of the vaccines). The horizontal axis (x) represents the transformation of the vaccine dose into Ln (Dose). The horizontal axis is plotted and constrained by the software according to the three dilutions of the vaccine used.

Point 10 : Lines 310-397 : the discussion should be shortened. The writing is lack conciseness Response 10 : Lines 310-397 : Thank you for your recommendation. we have tried to discuss and summarize the validation criteria of Kendrick's test point by point and to indicate the raised issues. All corrections have been made and highlighted through the track modifications style.

Point 11 : The MS should be improved by a through English proofreading.

Response 11 : Thank you for your recommendation . The manuscript was revised by Prof Touil Boukoffa Chafia, Prof Raache Rachida, Prof Zidi Ines and Prof Ouzari Hadda-Imene. Moreover, English language of the manuscript was edited by Prof Wahiba Kouki, English teacher at the Faculty of Sciences of Tunis (cited within the acknowledgment section). Spelling and grammar mistakes were revised throughout the manuscript, including some errors within the tables and figure legends (L22, L26, L39, L50, L68, L83, L84, L92, L95-96, L112, L144, L247-253, L393, L417, L433, L479,…….).

Reviewer 2 Report

In this article, the authors evaluate the potency of whole-cell pertussis vaccine by mouse protection test (MPT). After validating the infection model of B. pertussis, MPT is performed with immunization while the test vaccine and a standard vaccine. The results are compared by relative probit analysis to determine the potency of the test vaccine. The results are clearly presented and the variability of the MPT tests are extensively discussed. From five trials, only two can conclude to a good potency. The authors could discuss if five trials are enough or more trials are required for a more robust protocol for validating vaccine potency.

Minor typo errors:

  • UI is used sometimes instead of IU (line 28, 86, 107, 114, Table 3)
  • DL50 line 134
  • ‘diluant” in Table 1 for Trial 5

  • C.N. in legend of Table 2
  • “common slopes” are not visible for all graphs in Figure 1
  • fuducial in line 177, 303
  • paragraph 179-183 is duplicated from “Ethical considerations” section 2.7

Author Response

Response to Reviewer 2 Comments

We are very grateful to reviewer 2 for his thoughtful suggestions and recommendations, which greatly improved the quality of the manuscript. Based on your suggestions, we have made careful modifications to the manuscript. We would like to thank you for reviewing the manuscript and taking the time to improve our article.

Here we present our point-by-point response to the comments.

In this article, the authors evaluate the potency of whole-cell pertussis vaccine by mouse protection test (MPT). After validating the infection model of B. pertussis, MPT is performed with immunization while the test vaccine and a standard vaccine. The results are compared by relative probit analysis to determine the potency of the test vaccine. The results are clearly presented and the variability of the MPT tests are extensively discussed. From five trials, only two can conclude to a good potency. The authors could discuss if five trials are enough or more trials are required for a more robust protocol for validating vaccine potency.

We agree with Reviewer 2, that more tests are needed and that five tests are insufficient. Indeed a secondary validation is being programmed. Accordingly, we have signaled in the text the need to consider and schedule others trials to better establish validation protocols.

Minor typo errors:

Point 1 : UI is used sometimes instead of IU (line 28, 86, 107, 114, Table 3)

Response1 :

Line 28 : Thank you. We apologize .The IU unit is replaced by IU.

Line 86 and 107 :

For more details, we would like also mention that the unit of measurement indicated on the leaflet of the pertussis vaccine tested is the OU (Unit of Opacity). Futhermore, the unit of the standard vaccine is mentioned in IU. In the same way, the dilutions of each vaccine are carried out according to the manual quality control protocol recommended by the WHO for the pertussis vaccine. This part is detailed in the manuscript.

Line 114 : For clarification, 10 OU is the concentration used to adjust the opacity of the B. pertussus bacterial suspension used in the test (10 OU corresponds to 10x109 organisms/ml, using the 5th international opacity standard recommended by the WHO)

Table 3 : Thank you so much. Indeed this was a typing error. The unit IU is replaced by IU.

Point 2 : DL50 line 134

Response 2 : Line 134 : It is corrected. DL50 is replaced by LD50 (lethal dose median).

Point 3 : ‘diluant” in Table 1 for Trial 5

Response 3 : I apologize for the mistake. Table 1 for Trial 5. The word diluant was replaced by diluent.

Point 4 : C.N. in legend of Table 2

Response 4 : Thank you so much .Table 2, in legend , the C.N Abreviation is replaced by N.C.

Point 5 : “common slopes” are not visible for all graphs in Figure 1

Response 5 : It is corrected. Figure 2 « Common slopes » . The formatting is carried out for all the graphs.

Point 6 : fuducial in line 177, 303

Response 6 : Thank you so much.  The word « fuducial » is replaced by « fiducial » and , the FI abbreviation is corrected and replaced by FL

Point 7 : paragraph 179-183 is duplicated from “Ethical considerations” section 2.7

Response 7 : Line 179-183. Thank you so much .We acknowledge reviewer’ 2 s comment.  the paragraph is deleted because it is  duplicated from “Ethical considerations” section 2.7.

Reviewer 3 Report

The paper presents the assessment of anti-pertussis potency of whole cell pertussis vaccines using a test based on mice protection. In countries using whole-cell pertussis vaccines, the procedure could be used to assess the potency of these vaccines. 

I suggest to the authors the following points to increase the quality of the paper:

  1. A Figure presenting the phases of the MPT-based procedure could  support the explanations the text..
  2. Section 2.4. The procedure based on the MPT could be explained in more detail. Lines 103-110 are confusing. The procedure should be explained in terms of: phases of the procedure, requirements for validity of results, indicators used to assess immunogenicity activity.
  3. Lines 146-148. What data was used to evaluate the median immunizating dose ED50 and the relative potency of vaccines?
  4. Section 2.6. It seems that this section is related to the MPT procedure. It is not clear the connection between the first sentence indicating that at least 87.5% of mice immunized must survive and the following explanation. Is it necessary that at least 87.5%% of mice survive in addition with the criteria for validating a trial explained in lines 163-178?
  5. Line 239. “highest dilution”?. The percentage of immunized is lower than 87.5% for the 1.1 dilution.
  6. Lines 246-248. This sentence is confusing. Results presented in table 3 are not explained I n sufficient detail.
  7. Table 3. Why the potency obtained for the vaccine assessed in trial 1 is considered correct if the percentage of immunized mice was lower that 87.5% in 3 out of 6 assessments, as it is shown in Table 2?
  8. Lines 291-304. Based on criteria indicated on lines 175-177, only 2 vaccines passed the test, but it is not clear why results obtained in trial 1 could be used to obtain the combined potency.  

Round 2

Reviewer 1 Report

The manuscript has been much improvement.

Reviewer 3 Report

The revised version has improved the quality of the original paper.